# Aligning Out-of-Distribution Web Images and Caption Semantics via Evidential Learning

## ABSTRACT

Vision-language models, pre-trained on web-scale datasets, have the potential to greatly enhance the intelligence of web applications (e.g., search engines, chatbots, and art tools). Precisely, these models [15, 24] align disparate domains into a co-embedding space, achieving impressive *zero-shot* performance on multi-modal tasks (e.g., image-text retrieval, VQA). However, existing methods often rely on well-prepared data that less frequently contain noise and variability encountered in real-world scenarios, leading to severe performance drops in handling out-of-distribution (OOD) samples. This work first comprehensively analyzes the performance drop between in-distribution (ID) and OOD retrieval in Fig. 1. Based on the observations, this paper introduces a novel approach, Evidential Language-Image Posterior (ELIP) to achieve robust alignment between web images and semantic knowledge across various OOD cases by leveraging evidential uncertainties. The proposed ELIP can be seamlessly integrated into general image-text contrastive learning frameworks, providing an efficient fine-tuning approach without exacerbating the need for additional data. To validate the effectiveness of ELIP, we systematically design a series of OOD cases (e.g., image distortion, spelling errors, and a combination of both) on two benchmark datasets to mimic noisy data in real-world web applications. Our experimental results demonstrate that ELIP improves the performance and robustness of mainstream pre-trained vision-language models against OOD samples on image-text retrieval tasks.

## KEYWORDS

Vision-language modeling, uncertainty estimation, evidential learning

### ACM Reference Format:

Anonymous Author(s). 2024. Aligning Out-of-Distribution Web Images and Caption Semantics via Evidential Learning. In *Proceedings of Aligning Out-of-Distribution Web Images and Caption Semantics via Evidential Learning (Conference acronym 'XX)*. ACM, New York, NY, USA, 9 pages. https://doi.org/XXXXXXX.XXXXXXX

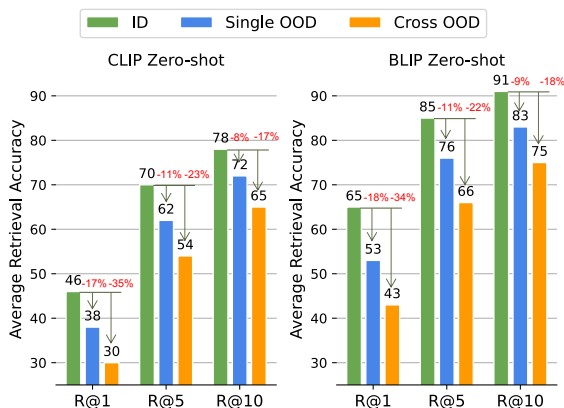

**Figure 1: Average performance in terms of Recall@K (R@K) among the image-text retrieval. To measure the vulnerability of large-scale pretraining (e.g., CLIP and BLIP) against OOD samples, we evaluate them under simple noisy cases (OOD-image: Gaussian noise, random rotate OOD-text: natural noise [21]). Also, we calculate the performance drop (MMI) [23] scores between the ID and OOD retrieval to test the robustness of each model.**

## 1 INTRODUCTION

Web applications such as search engines, recommendation systems, etc., greatly benefit human daily life [6, 8, 14, 28, 34], most dealing with complicated data formats from different domains (e.g., search engines require massive semantic knowledge, and recommendation

systems rely on image and text data). Among these web applications, multi-modal data of vision and language (VL) usually play an indispensable role [25], and have attracted remarkable research efforts [30, 31] in recent years. Particularly, CLIP [24] aligns vision and language domains into a shared embedding space, showing a promising zero-shot learning capacity for broad applications. However, web data frequently contend with many practical challenges, such as low-resolution images due to unreliable internet connections and text marred by garbled characters, leading to many out-of-distribution (OOD) samples compared with the clean, well-prepared training data. This gap raises a question – *will the pre-trained VL models be vulnerable to OOD samples in web applications?*

To investigate the above question, Fig. 1 shows an empirical study of two pre-trained VL models (CLIP [24] and BLIP [15]) for image and text retrieval over in-distribution (ID) and OOD samples. The clear performance drop between ID and OOD retrieval of these two state-of-the-art models inevitably casts a shadow of directly applying VL models to handle the wild web data. While fine-tuning the VL model with OOD data (varying with different domains) could be a solution, it is highly costly and generally infeasible due to the unknown data on the fly. Thus, we will propose an efficient uncertainty-aware fine-tuning approach to mitigate the negative impact of OOD samples on the pre-trained VL models.

Typically, there are three categories to uncertainty modeling: 1) deep ensemble [13], 2) variational inference [3, 4], and 3) deep evidential learning [1, 2, 26, 33]. Accounting for the large size of recent VL models, the first two uncertainty estimation methods may be less applicable since they both require multiple inference steps,

which can be computationally expensive, especially for the image-text ranking problem (where the pairwise calculation occurs). By contrast, deep evidential learning [5] provides explicit uncertainty representations based on a single forward pass, enriching uncertainty knowledge without additional inference costs. However, it is still under-explored in large-scale VL models since fine-tuning such networks requires high memory and computation requirements.

In this study, we fill in the gap of reasoning uncertainty for VL models by marrying deep evidential uncertainty into a parameter-efficient tuning framework. Concretely, we propose a novel Evidential Language-Image Posterior (ELIP) method, which leverages evidential learning with VL alignment to improve the generalization and reliability of pre-trained VL models in both ID and OOD cases. The proposed ELIP develops adapter [7, 10] layers to fine-tune the pre-trained VL models to acquire evidence knowledge by optimizing using evidential loss. Compared to traditional contrastive learning methods that primarily focus on point estimation for the class proba-bility of a sample, the evidential loss framework considers the entire probability distribution over all samples [26], enhancing robust-ness against OOD samples and disclosing less confident predictions. Based on the ID and OOD retrieval settings, we conduct extensive experiments to demonstrate the effectiveness of ELIP. Our method outperforms several state-of-the-art VL models on image-text re-trieval in most OOD cases. Our work showcases the potential of evidential learning for VL models and its importance in improving model reliability in realistic web scenarios. We summarize the main contributions of this work as follows.

- We introduce and design multiple OOD cases to investigate large-scale VL models against various noise on web data. We provide analysis of the MultiModal Impact (MMI) [23] score and uncertainty estimation based on ID and OOD samples, thoroughly discussing the robustness and reliability of VL models on image-text retrieval tasks.
- We propose a novel uncertainty-aware, parameter-efficient tuning method termed ELIP. The proposed ELIP adopts evidential learning to integrate image-text matching and uncertainty estimation in a single forward pass.
- Extensive experiments show that our method improves state-of-the-art VL models, CLIP [24] and BLIP [15], on image-text retrieval tasks against diverse OOD samples.

## 2 OUT-OF-DISTRIBUTION SCENARIOS

We introduce two OOD scenarios based on benchmark datasets (e.g., MS-COCO and FLickr30k), aiming to mimic diverse practi-cal web noisy data to assess the effectiveness of our approach and mainstream VL models for image-text retrieval tasks. We first in-troduce *simple OOD* cases by adding random Gaussian noise into each image with the normal distribution variance as 0.1 or subject-ing each image to a random rotation within 0 to 180 degrees. We use the same random seed in all experiments to ensure consistent generation for the random rotation. For textual input, we adopt the implementation described in [21], generating naturally noisy text encompassing various error aspects, including diacritics, casing, spelling, suffix/prefix alterations, punctuation variations, whitespace anomalies, word order shifts, insertions, and replacements. Notably,

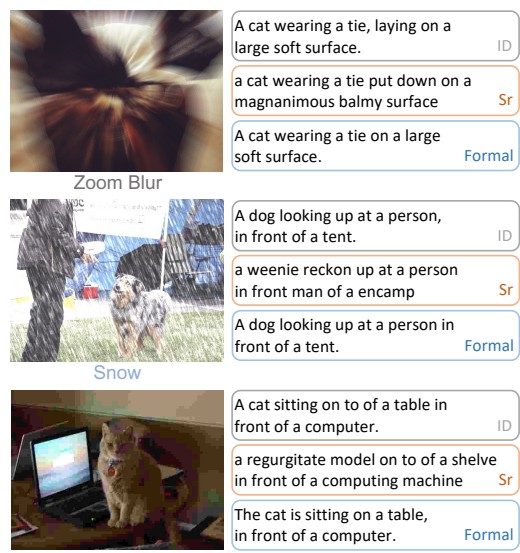

**Figure 2: Generated OOD web images and text for OOD re-trieval. We present web OOD images (e.g., zoom blur, snow, low resolution) paired with one ID and two web OOD texts (e.g., synonym replacement (sr) and formal [23]).**

these noisy samples are generated without the reliance on manually designed rules, enhancing the realism of the perturbations.

Secondly, we introduce ***web OOD*** cases (Fig. 2). In realistic web applications, massive amounts of low-quality images are uploaded to the web every day. Some common cases include non-focus im-ages, overexposed images, and compressed images. To mimic such noises, we follow [23] by utilizing blur (zoom), weather (snow), and compression (JPEG) as image-OOD perturbations. Also, the web contains a tremendous amount of noisy image description, which includes spelling and disordered issues. This paper uses word-level synonym replacement (sr) and sentence-level (formal) perturbation to generate noisy captions. We analyze encompass results aggregated across five perturbation levels for each type of web OOD case. This paper mainly focuses on testing the model's robustness against OOD cases. As shown in Fig. 3, we have 10% of simple OOD cases and 90% of web OOD cases over image and text domains.

## 3 METHODOLOGY

### 3.1 Overall Architecture

**Vision-language Contrastive Learning.** Recent vision-language (VL) models use vision transformer as the image encoder to encode an input image $I$ into a sequence of embeddings as $\{v_{cls}, v_1, \cdots, v_N\}$. They also employ a transformer network as the text encoder to trans-form input text $T$ into a sequence of embeddings $\{w_{sos}, w_1, \cdots, w_{eos}\}$. Where $v_{cls}$ and the activation of the highest layer of the transformer of $w_{eos}$ are treated as extracted features are normalized and linearly projected into a multi-modal $D$-dimension embedding space. We use $v \in \mathbb{R}^D$ and $w \in \mathbb{R}^D$ to denote the image and text features.

To learn an unimodal representation, image-text contrastive learn-ing is leveraged to learn a similarity function. Specifically, the image-to-text and text-to-image similarities between one query sample and

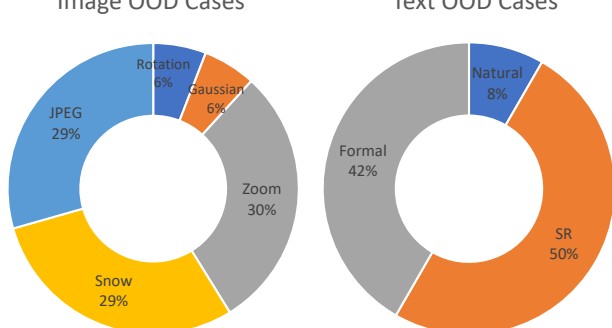

**Figure 3: Case study. Visualize the percentage of different OOD cases in the image and text domain. For simple OOD, we have rotation, Gaussian, and natural. For web OOD, we have snow, zoom, JPEG, formal, and synonym replacement (sr).**

all positive and negative samples in the target set were computed as:

$$\rho^{i2t} = \iota \left\langle v^\top W_0, \cdots, v^\top W_M \right\rangle,$$
$$\rho^{t2i} = \iota \left\langle w^\top V_0, \cdots, w^\top V_M \right\rangle, \qquad (1)$$

where $\iota$ is a logit-scale. $V \in \mathbb{R}^{M*D}$ and $W \in \mathbb{R}^{M*D}$ are image-text pairs representations. $\rho^{i2t}$ and $\rho^{t2i}$ can be used to find the correct matching in a top-K list (retrieval), such that parallel image-text pairs should return higher similarity scores. Let $y^{i2t}$ and $y^{t2i}$ be the one-hot label, representing positive sample as 1 and negative sample as 0. The image-text contrastive loss, consisting of image-to-text and text-to-image matching, is defined with cross-entropy ($\ell$) as

$$\mathcal{L}_{itc} = \frac{1}{2}[\ell(y^{i2t}, \sigma(\rho^{i2t})) + \ell(y^{t2i}, \sigma(\rho^{t2i}))], \qquad (2)$$

where $\sigma$ is a softmax function. However, Eq. (2) only considers the alignment between the correct pairs when getting the cross-embedding, without modeling the uncertainty between the query and all the other target samples. To estimate uncertainty in cross-alignment, this work introduces evidential knowledge in contrastive learning by learning a distribution over the similarities between all the cross-embedding.

**Bottleneck Adapter.** Adapter module [7] can be easily plugin-and-play into existing network to enable parameter-efficient transfer learning. Specifically, the adapter is a bottleneck structure with linear layers governed by a residual connection between the block's input and output. This work used the pre-trained CLIP [24] and BLIP [15] as the backbone models. Following the approach in [7], we inserted one adapter after the self-attention and MLP layers, respectively, in each transformer layer of the vision and language encoders (see Fig. 4). Eventually, the CLIP model has 64M trainable extra parameters, accounting for 13% of the entire model, while the BLIP model has 141M trainable extra parameters, which is 38%. We obtained new image and text features after passing through the pre-trained normalization and linear projection layers. These features were then used to compute the similarities $\rho^{i2t}$ and $\rho^{t2i}$ in Eq. (1).

## 3.2 Uncertainty Estimation with Cross Embedding

Recent evidential deep learning [1, 26] methods aim to overcome the limitations of the standard Softmax-based model for uncertainty estimation. Specifically, the Softmax function provides a point estimation for the matching similarity between the query and targets, which keeps reporting low uncertainty in OOD cases. Differently, the evidential deep learning framework models the uncertainty by placing a Dirichlet distribution (Dir) over the probability distribution. Also, deep evidential allows quantifying the uncertainty under a well-defined theoretical framework by leveraging Subjective Logic (SL) [9]. Typically, SL is beneficial when there are multiple sources of information with varying levels of trustworthiness or when dealing with subjective opinions and beliefs. In this paper, the proposed method targets an image-text retrieval task, which involves feature alignment and a ranking process that contains multiple sources of information and different levels of trustworthiness, respectively. Therefore, we consider using Subjective Logic to quantify cross-modal retrieval uncertainty.

Typically, SL considers a frame of K mutually exclusive singletons (e.g., class labels) by providing a belief mass

$$b_k = \frac{e_k}{\sum_{i=1}^{K}(e_i + 1)}, \qquad (3)$$

for each singleton $k = 1, \cdots, K$, where $e_k > 0$ is the evidence derived for the $k^{th}$ singleton. Note that the overall uncertainty mass of $u$ and all non-negative belief masses are sums up to one, i.e.,

$$u = 1 - \sum_{k=1}^{K} b_k = \frac{K}{\sum_{i=1}^{K}(e_i + 1)}, \qquad (4)$$

where the uncertainty is also inversely proportional to the total evidence. When the evidence for each singleton is zero, the total belief is zero, and the uncertainty is one. Current methods have different theories to define the Dir. Generally, the evidence assigned corresponds to a Dir with parameters $\alpha_k = e_k + 1$. While in another work [26], given a sample $x_k$ and a classifier $f(\theta)$ with parameters $\theta$, the corresponding Dir has parameters $\alpha_k = f(x_k \mid \theta) + 1$.

However, this work considers cross-domain information, which differs from the previous methods that use single-domain data. Specifically, we use multi-modal embedding and define $\alpha$ using cross similarities between $M$ image-text pairs. Therefore, the subject opinion for the $i^{th}$ query and the $j^{th}$ target sample can be computed from the parameters of the corresponding Dir using

$$b_j^{(i)} = \frac{\alpha_j^{(i)} - 1}{\sum_{l=1}^{M}(\alpha_l^{(i)})}. \qquad (5)$$

Let $\alpha^{(i)} = < \alpha_1^{(i)}, \cdots, \alpha_M^{(i)} >$ become the parameter of a Dirn for the cross similarities, then $(\alpha_j^{(i)} - 1)$ is the evidence estimated by the matching similarity between the $i^{th}$ query and the $j^{th}$ target sample, where $i, j = 1, \cdots, M$. Finally, given these parameters, the prediction uncertainty can be computed using Eq. (4) for each query samples.

Specifically, we define evidence as a measure of the amount of similarity between query and target samples in favor of aligning the positive sample and pushing away the negative samples. For convenience, we assign the similarity vector $\rho \in \mathbb{R}^{M*M}$ computed in Eq. (1) as the general representation for $\rho^{i2t}$ and $\rho^{t2i}$, since image-to-text and text-to-image similarities share the same computation process for evidence. Also, we assign $\alpha$ to represent $\alpha^{i2t}$ and $\alpha^{t2i}$. This work defines the Dir over cross-embedding between the query and the target samples. By taking the cross similarities $\rho^i \in \mathbb{R}^M$

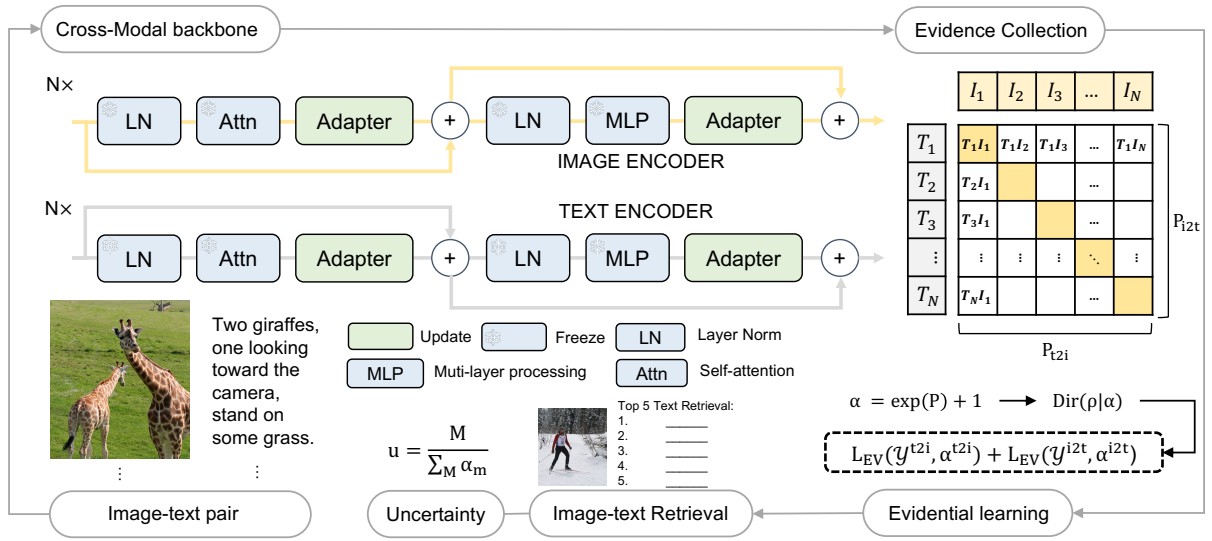

**Figure 4: Illustration of our proposed ELIP model. Our method can perform image-text retrieval and uncertainty estimation in a single forward step. The image and text encoder were fine-tuned by different adapters with scalable parameters on clean data without augmentation. We develop a new evidential loss ($\mathcal{L}_{ev}$) to implement image-text matching tasks, and the learned Dirichlet distribution (Dir) posterior is used for uncertainty estimation and OOD detection.**

between the $i^{th}$ query and all target samples, the $j^{th}$ parameter of the Dir $\alpha_j^{(i)}$ is computed as

$$\alpha_j^{(i)} = \exp\left(\rho_j^{(i)}\right) + 1. \tag{6}$$

We apply $\exp(\cdot)$ as an activation function to ensure positive evidence for all cross-embedding. Because $\rho^{(i)}$ is the cross similarity between image and texts, the value is greater than zero only for the parallel pair. Eventually, Eq. (6) takes input computed in Eq. (1), and the output $\alpha$ can be used to calculate uncertainty in Eq. (4). Our proposed $\alpha$ surprisingly connects cross-modal alignment and evidential learning in a single forward pass.

Eventually, our model learns and updates the Dir by using the image-text similarity as subjective opinions and collects evidence that leads to those opinions. During training, the expected matching similarity for the $i^{th}$ query and the $j^{th}$ target sample is computed as

$$\mathbb{E}[p_j^{(i)}] = \frac{\alpha_j^{(i)}}{\sum_{l=1}^{M} \alpha_l^{(i)}}, \tag{7}$$

since the distribution is a probability density for possible values of the probability mass $p$, where $p_j^{(i)} \in [0, 1]$. For convenience, we assign $p$ to represent $p^{i2t}$ and $p^{t2i}$. Throughout the training process, when an observation about a query sample relates it to one of the $M$ target samples, the corresponding Dirichlet parameter is incremented to update the Dirn with the new observation. For instance, the increment matching similarity between image and text may contribute to its feature alignment, which may benefit image/text encoder learning.

## 3.3 Learning with Evidential Knowledge

Having formalized the use of a Dirichlet Distribution to capture evidence knowledge, we next describe our approach for optimizing the model to output the parameters of this distribution. For the VL model, the objective is to align two domains into the same space, and we follow this idea by first calculating the similarity between the image and text features. Secondly, instead of using the matching score for gradient computation, we structure the learning process with two distinct parts: (1) acquiring model evidence to support our observation and (2) minimizing evidence uncertainty when the feature alignment is low. Eventually, we can fit our data to the evidential model at a high level while enforcing a prior to remove false evidence and inflate uncertainty.

**Evidential Loss.** To better explain, we assign $\alpha$ to $\alpha^{(i)}$ as the cross similarities between the $i^{th}$ query and all target samples in the following sections. We define a loss function and compute its Bayes risk for the Dirichlet parameters. For image-to-text and text-to-image matching, we denote the Bayes risk as

$$
\begin{aligned}
\mathcal{L}^{i2t} &= \sum_{j=1}^{M} y_j^{i2t} (\psi(S^{i2t}) - \psi(\alpha_j^{i2t})), \\
\mathcal{L}^{t2i} &= \sum_{j=1}^{M} y_j^{t2i} (\psi(S^{t2i}) - \psi(\alpha_j^{t2i})),
\end{aligned}
\tag{8}
$$

where $\psi(\cdot)$ is the *digamma* function and $S = \sum_{j=1}^{M} \alpha_j$ is Dirichlet strength, we assign $S$ to represent $S^{i2t}$ and $S^{t2i}$.

**Minimizing Evidence on Errors.** The evidential loss provides an objective function for the training model to output image and text feature alignment distribution to fit the observations by maximizing the model evidence. However, due to the negative samples in the training batch, the model may be misdirected and put strong evidence for the wrong prediction. Therefore, we describe how to regularize training by applying an incorrect evidence penalty and aim to minimize the evidence of incorrect matching. We define $\tilde{a} = y + (1 - y) \odot a$, where $\tilde{a}$ and $y$ represent $\tilde{a}^{i2t}$, $\tilde{a}^{t2i}$ and $y^{i2t}$, $y^{t2i}$. Consequently, we incorporate a Kullback-Leibler (KL) divergence term into our loss function, where KL can be a regularization term by penalizing those divergences from negative samples that do not contribute to data fit. Overall, the evidential loss $\mathcal{L}_{ev}(\theta)$ consists of

**Table 1: Comparison of performance in terms of Recall@K (R@K) and MMI [23] score among ID and simple-OOD retrieval. CLIP and BLIP are pre-trained *zero-short*; the others were fined-tuned on clean MS-COCO. ELIP and ELIP+ were transfer learned from pre-trained CLIP and BLIP. For ID image (I) and ID text (T) retrieval, BLIP reports the best performance, but ELIP surpasses other models in retrieval between OOD Image (I*) and OOD text (T*). From the MMI score, ELIP achieves the lowest performance drop.**

| Image Retrieval | T → I | | | T → I* | | | T* → I | | | T* → I* | | | MMI | | |
|---|---|---|---|---|---|---|---|---|---|---|---|---|---|---|---|
| | R@1 | R@5 | R@10 | R@1 | R@5 | R@10 | R@1 | R@5 | R@10 | R@1 | R@5 | R@10 | R@1 | R@5 | R@10 |
| CLIP [24] | 35.3 | 60.0 | 70.2 | 30.4 | 54.4 | 65.3 | 27.7 | 50.8 | 61.3 | 24.2 | 46.4 | 56.9 | ↓22.3% | ↓15.8% | ↓12.9% |
| BLIP [15] | 56.9 | 80.8 | 87.9 | 43.1 | 67.8 | 76.5 | 50.0 | 74.7 | 82.8 | 36.9 | 60.6 | 70.1 | ↓23.8% | ↓16.2% | ↓13.0% |
| ALBEF [16] | 60.7 | 84.3 | 90.5 | 47.8 | 72.0 | 80.3 | 51.9 | 76.8 | 85.6 | 41.2 | 65.6 | 74.7 | ↓22.6% | ↓15.2% | ↓11.4% |
| BLIP [15] | **64.3** | **85.7** | **91.5** | 51.4 | 74.5 | 82.1 | **57.2** | **80.3** | **87.4** | 45.2 | 68.8 | 77.2 | ↓20.3% | ↓13.0% | ↓10.1% |
| ELIP (ours) | 60.4 | 83.9 | 90.5 | **51.5** | **76.9** | **85.0** | 52.3 | 76.9 | 85.1 | 43.7 | **69.4** | **78.8** | **↓18.6%** | **↓11.3%** | **↓8.3%** |
| ELIP+ (ours) | 63.7 | 85.4 | 91.3 | 51.0 | 74.5 | 82.3 | 57.0 | 80.0 | 87.2 | **45.6** | 69.3 | 77.8 | ↓19.6% | ↓12.6% | ↓9.7% |

| Text Retrieval | I → T | | | I* → T | | | I → T* | | | I* → T* | | | MMI | | |
|---|---|---|---|---|---|---|---|---|---|---|---|---|---|---|---|
| | R@1 | R@5 | R@10 | R@1 | R@5 | R@10 | R@1 | R@5 | R@10 | R@1 | R@5 | R@10 | R@1 | R@5 | R@10 |
| CLIP [24] | 56.0 | 79.6 | 86.9 | 46.3 | 71.1 | 79.9 | 46.1 | 71.5 | 80.5 | 36.6 | 62.5 | 73.0 | ↓23.3% | ↓26.7% | ↓10.5% |
| BLIP [15] | 72.5 | 90.0 | 94.7 | 52.1 | 73.4 | 81.0 | 67.6 | 87.9 | 93.3 | 48.2 | 71.1 | 78.9 | ↓22.8% | ↓13.9% | ↓10.9% |
| ALBEF [16] | 77.6 | 94.3 | 97.2 | 59.8 | 79.5 | 85.3 | 71.0 | 90.6 | 94.9 | 54.7 | 75.7 | 82.4 | ↓20.3% | ↓13.1% | ↓9.9% |
| BLIP [15] | **81.9** | **95.4** | **97.8** | 64.8 | 82.6 | 87.6 | **76.4** | **93.3** | **96.5** | 59.8 | 79.5 | 85.5 | ↓18.2% | ↓10.8% | ↓8.1% |
| ELIP (ours) | 77.5 | 94.2 | 97.0 | **66.3** | **86.0** | **91.7** | 71.3 | 90.8 | 95.0 | **60.0** | **82.2** | **88.7** | **↓15.0%** | **↓8.4%** | **↓5.4%** |
| ELIP+ (ours) | 81.3 | 95.2 | 97.7 | 64.6 | 82.6 | 87.8 | 76.2 | 92.9 | 96.2 | 59.9 | 79.6 | 85.4 | ↓17.7% | ↓10.7% | ↓8.1% |

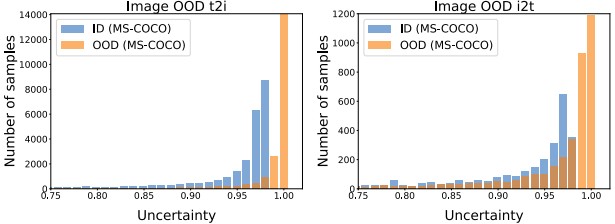

**Figure 5: OOD detection by uncertainty of ELIP on ID and OOD image-text retrieval on MS-COCO. The uncertainty values are in the range (0.75–1.00) within each distribution.**

two terms for maximizing and regularizing evidence, scaled by $\lambda_t$

$$\mathcal{L}_{ev}^{i2t} = \mathcal{L}^{i2t} + \lambda_t \text{KL}[D(p^{i2t}|\tilde{\alpha}^{i2t})||D(p^{i2t}|\langle 1, \cdots, 1 \rangle)],$$
$$\mathcal{L}_{ev}^{t2i} = \mathcal{L}^{t2i} + \lambda_t \text{KL}[D(p^{t2i}|\tilde{\alpha}^{t2i})||D(p^{t2i}|\langle 1, \cdots, 1 \rangle)], \quad (9)$$

where $\lambda_t = min(1.0, t/15)$ is the annealing coefficient, t is the index of the current training epoch, $D(p|\langle 1, \cdots, 1 \rangle)$ is the uniform Dirichlet distribution, and $\tilde{\alpha}$ is the Dirichlet parameters of misleading evidence from $\alpha$. The KL divergence term $KL[D(p|\tilde{\alpha})||D(p|\langle 1, \cdots, 1 \rangle)]$ can be compute as

$$\log\left(\frac{\Gamma(\tilde{S})}{\Gamma(M)\prod_{j=1}^{M}\Gamma(\tilde{\alpha}_j)}\right) + \sum_{j=1}^{M}(\tilde{\alpha}_j - 1)[\psi(\tilde{\alpha}_j) - \psi(\tilde{S})].$$

We use dynamic scaling $\lambda_t$ to modify the weights of $KL$ term, leading the model to focus on learning relationships between positive pairs at the beginning and gradually put more attention on negative pairs. Specifically, by gradually increasing the effect of the $KL$ divergence, we allow the neural network to explore the parameter space and avoid premature convergence to the uniform distribution for the misaligned samples.

Empirically, the total loss $\mathcal{L}_{EV}$ consists of two terms to update the image and text encoder evenly:

$$\mathcal{L}_{ev} = \frac{1}{2}(\mathcal{L}_{ev}^{i2t} + \mathcal{L}_{ev}^{t2i}). \quad (10)$$

Overall, we leverage evidential loss to fine-tune the pre-trained CLIP and BLIP models using ID data. By updating the inserted adapters, ELIP can preserve high performance on ID retrieval tasks while achieving reliable performance on OOD retrieval tasks (refer to Table 1). During training within a high-level embedding dimension, the model captures deeper connections between images and text, which enables the generation of evidence for pairwise feature alignment based on these patterns, thereby minimizing the overall loss.

**Datasets and Evaluation Metrics.** We train and evaluate our model on MS-COCO [18] and Flickr30K dataset [32]. We follow the splits of COCO-Karpathy, which contains 118,287 image-text pairs for training and 5000 for testing. Flickr30K consists of 31783 image-text pairs, and we use the standard training and test split [11], which contains 28000 and 1000 samples. We evaluate the performance of our model using the common Recall@K (R@K) metric, which measures the proportion of correct matches among the top K retrieved results. Based on our OOD cases, Table 1 illustrates five evaluation metrics. The examples of image retrieval: R@K over ID retrieval (T → I), text-OOD retrieval (T* → I), image-OOD retrieval (T→ I*), multi-OOD retrieval (T* → I*), and MultiModal Impact score (MMI) [23] (% of performance drop between ID and OOD retrieval).

**Implementation Details.** We use pre-trained CLIP *zero-shot* and BLIP *fine-tuning* as our backbone models and initialize our implementation with their weights. To fine-tune the model efficiently, we independently modify the image and text encoder by inserting adapters. Expressly, we set the bottle-neck feature dimension to half of the feature dimension from the previous layer, and we use RELU as the activation function. In order to sustain the performance pre-trained on previous knowledge, we initialize all new parameters of adapters with values drawn from the normal distribution with $\mu = 0$, and $\sigma = 0.001$. We fine-tune all the models for 30 epochs with a batch size 280. We use the AdamW [19] optimizer with an initial

**Table 2: Comparisons of MMI [23] scores between three models in OOD retrieval. We utilize five web OOD cases generated from MS-COCO, including OOD-image (zoom blur, snow noise, JPEG compression) and OOD-text (synonym replacement (sr), formal). After fine-tuning on clean MS-COCO, ELIP achieves the lowest performance drop in most OOD retrieval cases, indicating better robustness against web OOD samples.**

| Image Retrieval | MMI by $I^*_{zoom}$ | | | MMI by $I^*_{snow}$ | | | MMI by $I^*_{JPEG}$ | | | MMI by $T^*_{sr}$ | | | MMI by $T^*_{formal}$ | | |
|---|---|---|---|---|---|---|---|---|---|---|---|---|---|---|---|
| | R@1 | R@5 | R@10 | R@1 | R@5 | R@10 | R@1 | R@5 | R@10 | R@1 | R@5 | R@10 | R@1 | R@5 | R@10 |
| ALBEF [16] | ↓51.9% | ↓39.1% | ↓32.7% | ↓26.0% | ↓15.8% | ↓11.7% | ↓8.9% | ↓5.1% | ↓3.4% | ↓13.7% | ↓7.8% | ↓5.5% | ↓0.8% | ↓0.5% | ↓**0.2%** |
| BLIP [15] | ↓50.5% | ↓37.7% | ↓31.7% | ↓22.7% | ↓13.1% | ↓9.5% | ↓6.5% | ↓3.2% | ↓2.2% | ↓13.7% | ↓7.2% | ↓5.2% | ↓1.2% | ↓0.5% | ↓0.3% |
| ELIP (ours) | ↓**32.7%** | ↓**20.7%** | ↓**15.8%** | ↓**13.0%** | ↓**6.6%** | ↓**4.0%** | ↓**2.5%** | ↓**1.7%** | ↓**1.0%** | ↓**7.0%** | ↓**4.6%** | ↓**3.3%** | ↓**0.5%** | ↓**0.5%** | ↓0.4% |
| Text Retrieval | R@1 | R@5 | R@10 | R@1 | R@5 | R@10 | R@1 | R@5 | R@10 | R@1 | R@5 | R@10 | R@1 | R@5 | R@10 |
| ALBEF [16] | ↓62.1% | ↓45.8% | ↓38.1% | ↓33.9% | ↓18.6% | ↓12.8% | ↓7.6% | ↓3.4% | ↓1.9% | ↓9.7% | ↓3.9% | ↓2.2% | ↓0.0% | ↓0.2% | ↓0.2% |
| BLIP [15] | ↓62.5% | ↓45.3% | ↓37.6% | ↓28.8% | ↓15.6% | ↓10.9% | ↓5.4% | ↓2.3% | ↓1.4% | ↓9.4% | ↓3.1% | ↓1.7% | ↓0.2% | ↓**0.2%** | ↓0.2% |
| ELIP (ours) | ↓**47.9%** | ↓**29.4%** | ↓**22.5%** | ↓**22.2%** | ↓**9.9%** | ↓**6.1%** | ↓**1.4%** | ↓**1.2%** | ↓**0.6%** | ↓**5.8%** | ↓**2.7%** | ↓**1.1%** | ↓**0.0%** | ↓0.3% | ↓**0.0%** |

learning rate of 5e-5, and the weight decayed with a rate of 0.02 for all the experiments.

## 4 EXPERIMENTS

### 4.1 Evaluation on Image-Text Retrieval

**MS-COCO.** We provide experimental results in two groups (simple OOD and web OOD). Table 1 provides image-text retrieval and MultiModal Impact score (MMI) [23] score under simple OOD cases. Notably, despite having fewer trainable parameters, ELIP outperforms the previous model, ALBEF, in most OOD settings. Even when benchmarked against the more robust backbone model BLIP fine-tuning, ELIP achieves superior performance on image OOD and cross-modal OOD settings for image and text retrieval. MMI score measures the relative performance drop between ID and OOD retrieval, providing a fair and objective assessment of the model's robustness facing OOD samples. As evident in the results, both ELIP and ELIP+ outperform all baseline models on the MMI benchmark, underscoring the efficacy of our approach in simple OOD scenarios.

In Table 2, we conduct an analysis of ELIP and other baseline models under web OOD cases. Following [23], we leverage zoom blur, snow noise, and JPEG compression in the vision domain and sr and formal in the language domain, which are commonly encountered in real-world web applications. The observations reveal that ELIP consistently outperforms all other baseline models in the context of image-text retrieval tasks by measuring the MMI score.

After experiments on MS-COCO, we provide some detailed analysis: 1) We notice that ELIP and ELIP+ perform better after fine-tuning with larger batch size, as they allow the model to learn from a more diverse set of negative samples, leading to a better understanding of complex evidence distributions. 2) Although ELIP+ may exhibit slightly worse performance than BLIP fine-tuning in some ID and OOD cases, this is expected since ELIP+ simplifies the training process used in BLIP. However, ELIP+ can provide more reliable uncertainty estimates than BLIP. Specifically, ELIP+ returns high uncertainty values for OOD retrieval cases, while BLIP is overconfident with noisy retrievals. 3) We found that ELIP can capture reliable similarities between OOD images and OOD text. Specifically, as shown in Fig.7, when all inputs are OOD, ELIP can return more

accurate retrieval results than ELIP w/o EV based on limited information. However, when image and text are highly damaged without helpful information, the top 1 retrieval will be significantly affected.

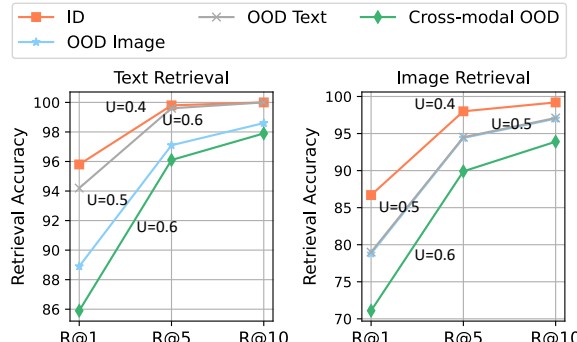

**Figure 6: ID and OOD image-text retrieval and uncertainty ($U$) estimation of ELIP on Flickr30K.**

**Flickr30k.** Given the considerable improvement of ELIP in OOD retrieval on MS-COCO, we further our study on Flickr30K. As shown in Table 3, ELIP outperforms most baseline models on the simple-OOD retrieval tasks. Also, ELIP and ELIP+ have the smallest performance drop between ID and OOD retrieval. Interestingly, we found that ELIP improves the pre-trained model more than ELIP+. This may be attributed to two factors: 1) the pre-trained CLIP constructs a better cross-embedding than BLIP, and 2) the pre-trained BLIP is already strong enough and nearly converges to the global optimum on the image-text retrieval tasks. Additionally, the comparison between ELIP and ELIP w/o EV demonstrates the effectiveness of evidential loss, enabling ELIP to achieve more reliable OOD image-text retrieval.

### 4.2 OOD Detection

ELIP demonstrates an ability to discern between ID and OOD retrieval by using uncertainty as a scoring criterion. As illustrated in Fig. 5, ELIP exhibits a noteworthy ability to identify anomalous retrieval outcomes when both query and target samples fall within the OOD category. This capability becomes apparent as the estimated uncertainties for OOD image-text retrieval results converge towards a value of 1.0 following the application of evidential

**Table 3: Comparison of performance in terms of Recall@K (R@K) and MMI [23] score among ID and simple-OOD retrieval. CLIP and BLIP are pre-trained *zero-short*, and the others were fined-tuned on clean Flickr30K. ELIP and ELIP+ were transfer learned from pre-trained CLIP and BLIP. I: ID image, I\*: OOD image, T: ID text, and T\*: OOD text.**

| Image Retrieval | T → I | | | T → I* | | | T* → I | | | T* → I* | | | MMI | | |
|---|---|---|---|---|---|---|---|---|---|---|---|---|---|---|---|
| | R@1 | R@5 | R@10 | R@1 | R@5 | R@10 | R@1 | R@5 | R@10 | R@1 | R@5 | R@10 | R@1 | R@5 | R@10 |
| CLIP [24] | 64.5 | 86.7 | 92.2 | 58.0 | 82.8 | 89.2 | 53.6 | 79.0 | 85.6 | 48.1 | 74.2 | 81.5 | ↓17.5% | ↓9.3% | ↓7.3% |
| BLIP [15] | 78.2 | 94.0 | 96.8 | 61.0 | 81.2 | 87.1 | 71.3 | 90.0 | 93.8 | 54.6 | 75.9 | 82.6 | ↓20.3% | ↓12.4% | ↓9.3% |
| ALBEF [16] | 85.5 | 97.5 | 98.9 | 68.8 | 86.6 | 91.0 | 78.6 | 94.4 | 96.8 | 62.4 | 82.2 | 87.7 | ↓18.2% | ↓10.0% | ↓7.1% |
| BLIP [15] | **87.3** | 97.6 | 98.9 | 72.3 | 89.0 | 92.8 | 78.2 | 94.0 | 96.8 | 61.0 | 81.2 | 87.1 | ↓19.2% | ↓9.8% | ↓6.7% |
| ELIP w/o EV | 85.3 | 97.9 | 99.0 | 78.3 | 94.3 | 97.0 | 78.2 | 94.2 | 96.9 | 70.7 | 89.1 | 93.3 | ↓**11.2%** | ↓5.5% | ↓3.3% |
| ELIP (ours) | 86.7 | **98.0** | **99.2** | **78.8** | 94.4 | 97.0 | 79.0 | **94.5** | **97.1** | **71.1** | **89.9** | **93.9** | ↓12.0% | ↓5.2% | ↓3.2% |
| ELIP+ (ours) | 86.5 | 97.1 | 98.3 | 78.0 | **94.6** | **97.4** | **80.4** | 94.2 | 96.3 | 70.0 | 89.5 | 93.3 | ↓12.0% | ↓**4.5%** | ↓**2.7%** |

| Text Retrieval | I → T | | | I* → T | | | I → T* | | | I* → T* | | | MMI | | |
|---|---|---|---|---|---|---|---|---|---|---|---|---|---|---|---|
| | R@1 | R@5 | R@10 | R@1 | R@5 | R@10 | R@1 | R@5 | R@10 | R@1 | R@5 | R@10 | R@1 | R@5 | R@10 |
| CLIP [24] | 84.3 | 97.9 | 99.3 | 76.0 | 93.7 | 96.6 | 76.4 | 94.5 | 97.5 | 67.1 | 90.0 | 93.7 | ↓13.2% | ↓5.3% | ↓3.4% |
| BLIP [15] | 87.4 | 98.1 | 99.2 | 69.1 | 85.4 | 89.9 | 85.8 | 97.6 | 98.7 | 66.4 | 85.7 | 89.8 | ↓15.6% | ↓8.7% | ↓6.5% |
| ALBEF [16] | 95.9 | 99.8 | 100.0 | 77.2 | 89.3 | 91.9 | 92.4 | 99.7 | 99.9 | 73.9 | 87.5 | 90.3 | ↓15.4% | ↓7.6% | ↓6.0% |
| BLIP [15] | **97.2** | 99.9 | 100.0 | 81.6 | 92.5 | 94.8 | 87.4 | 98.1 | 99.2 | 69.1 | 85.4 | 89.9 | ↓13.0% | ↓7.9% | ↓5.4% |
| ELIP w/o EV | 96.4 | 99.8 | 99.9 | 88.7 | 96.6 | 98.6 | 91.8 | 99.5 | 100.0 | 84.5 | 95.4 | 97.0 | ↓8.4% | ↓2.6% | ↓1.4% |
| ELIP (ours) | 95.8 | 99.8 | **100.0** | **88.9** | **97.1** | **98.6** | **94.2** | 99.6 | **100.0** | **85.9** | **96.1** | **97.9** | ↓**6.4%** | ↓**2.2%** | ↓**1.2%** |
| ELIP+ (ours) | 96.2 | **99.9** | 100.0 | 87.9 | 96.8 | 98.3 | 93.9 | **99.7** | 100.0 | 84.3 | 95.1 | 96.7 | ↓7.8% | ↓2.7% | ↓1.7% |

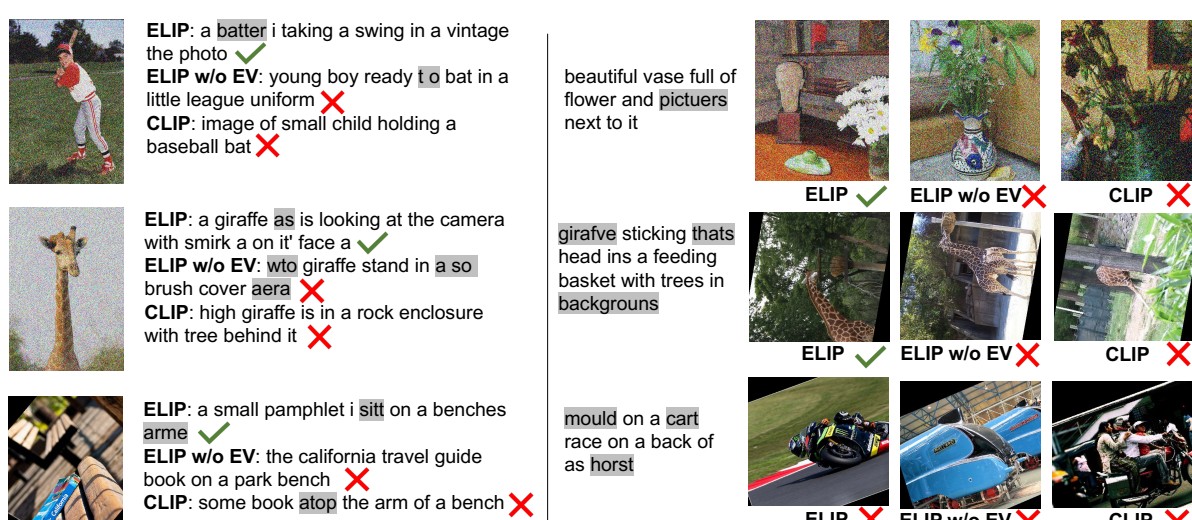

**Figure 7: Top 1 cross-domain OOD retrieval (OOD-image: Gaussian noise, random rotate OOD-text: natural noise) of three models on MS-COCO. Left: text retrieval. Right: image retrieval.**

learning. Conversely, during ID retrieval tasks, ELIP consistently furnishes meaningful uncertainty estimations, where the majority of ID retrieval instances yield uncertainties below the threshold of 0.8 since retrieval task involves much more complex Dirichlet distribution (larger class space) than regular classification tasks, which makes the distribution of uncertainty closer to relative large value. Consequently, ELIP emerges as a reliable uncertainty estimation method, particularly when confronted with OOD problems within the image-text retrieval tasks. Furthermore, we evaluate ELIP on Flickr30k (Fig. 6). Specifically, the estimated uncertainty is close to the threshold of 0.4 in the ID setting and surpasses the threshold

of 0.4 in OOD settings, which proves that ELIP has higher confidence for clean retrieval and lower for OOD retrieval. Overall, ELIP acquires a reliable uncertainty estimation and OOD detection ability.

### 4.3 Ablation Study

In Table 4, we investigate the impact of each component. We compare four strategies: 1) ELIP without adapters, 2) ELIP without image adapters, 3) ELIP without text adapters, and 4) ELIP without evidential learning. After fine-tuning on the same data and using the consistent pre-trained weights, we first observed a direct improvement when only the image adapters were applied. The observation indicates that the pre-trained vision encoder acquires better semantic

**Table 4: Ablation study of the proposed ELIP by Recall @ 1. We report ID retrieval and average results on the simple-OOD retrieval. For fine-tuning the projection layers of CLIP using evidential loss (ELIP w/o A), the performance of ID (I, T) retrieval and OOD ($I*$, $T^*$) retrieval is lower than fine-tuning with text adapter (ELIP w/o IA), fine-tuning with image adapter (ELIP w/o TA), ours (ELIP). Furthermore, the improvement of MMI [23] score between fine-tuning without evidence loss (ELIP w/o Ev) and ours (ELIP) proves the effectiveness of our method against the OOD issue.**

| Method | $I \rightarrow T$ | $T \rightarrow I$ | $I^* \rightarrow T$ | $T \rightarrow I^*$ | $I \rightarrow T^*$ | $T^* \rightarrow I$ | $I^* \rightarrow T^*$ | $T^* \rightarrow I^*$ | i2t MMI | t2i MMI |
|---|---|---|---|---|---|---|---|---|---|---|
| ELIP w/o A | 60.2 | 44.5 | 51.7 | 38.4 | 49.8 | 36.1 | 43.1 | 30.6 | ↓19.9% | ↓21.3% |
| ELIP w/o IA | 71.3 | 52.8 | 62.1 | 45.6 | 63.8 | 44.3 | 55.1 | 38.1 | ↓15.4% | ↓19.2% |
| ELIP w/o TA | 76.6 | 60.1 | 63.8 | 51.0 | 68.0 | 51.5 | 55.6 | 42.3 | ↓18.5% | ↓19.7% |
| ELIP w/o Ev | 76.7 | 60.3 | 64.3 | 51.4 | 70.5 | 51.9 | 58.2 | 43.3 | ↓16.1% | ↓19.0% |
| ELIP | **77.5** | **60.4** | **66.3** | **51.5** | **71.3** | **52.3** | **60.0** | **43.7** | ↓**15.0**% | ↓**18.6**% |

knowledge extraction with the assistance of extra adapters, leading to better cross-modal alignment. Furthermore, the model becomes more robust after optimizing using evidential learning since the MMI score of ELIP drops compared to ELIP w/o Ev. When all components were utilized, the effects of the adapters and evidential learning complemented each other, resulting in substantial improvements compared to regular image-text contrastive learning.

## 5 RELATED WORK

**Vision-language Modeling and its Web Application.** The current research focuses on vision-language (VL) pre-training, whether using an encoder-based or complex encoder-decoder-based structure. Encoder-based methods are mainly single-stream and double-stream methods, where single-stream uses a single transformer encoder to concatenate image and text embedding, e.g., VL-BERT [27], Image-BERT [22], Unified VLP [35], ViLBERT [20], and VisualBERT [17]. In comparison, double-stream methods use image and text encoders to extract features separately, e.g., CLIP. Some encoder-decoder-based models leverage cross-modal attention and combine multi-tasks (e.g., image-text retrieval, image captioning) to achieve better performance and higher flexibility on many downstream tasks, e.g., BLIP. In the meantime, due to the demand for large-scale data and the limitation of human-annotated data, most methods use image-data pairs collected from the Web like LAION [25], VG [12]. In our task, we exploit CLIP, a two-stream method with excellent image-text matching performance. As a significant step towards flexible and practical *zero-shot* classifier, CLIP has a clean and relatively simple structure, with two transformer networks used to extract the image features and text features respectively and finally cross-connect during loss calculation. Also, CLIP was trained using 400M image-text pairs collected from the Web. According to the results reported in CLIP, its image-text retrieval *zero-shot* performance surpasses some fine-tuned models. Due to this impressive performance, many works leverage the power of large-scale VL pre-training and benefit the development of web applications [31]. Therefore, powerful vision-language pre-training plays a significant role in recent web application studies.

**Uncertainty Estimation.** Recent studies declared that uncertainty estimation in DNN contains four different steps [5], (1) Data acquisition; (2) DNN building; (3) applied inference model; and (4) Prediction's uncertainty model. Several factors may cause model and data uncertainty and affect the model prediction. There are

many methods to achieve uncertainty estimation. Single deterministic methods predict uncertainty based on the forward pass; Bayesian methods consider the class probabilities and distributions [3, 4]. Evidential Deep Learning (EDL) [29] starts to attract attention due to its convenience, allowing the uncertainty computation to be done in a single forward pass and set of weights. Existing works show the benefits from EDL when doing uncertainty estimation on their model, including regression task [1, 2], classification task [26, 33]. We apply the EDL framework to the retrieval task, and the experimental results also show promising results for uncertainty estimation.

## 6 CONCLUSIONS AND SOCIAL IMPACTS

This paper introduces an innovative methodology aimed at harnessing the power of evidential learning within the context of noisy web images and semantic knowledge alignment. By preliminary analysis of the performance drop between ID and OOD retrieval, we propose ELIP to effectively against noisy samples. Intuitively, our work can benefit cross-modal related web applications such as robust retrieval and recommendation. To accomplish this, we employ adapters to facilitate the efficient fine-tuning of CLIP and BLIP. We progressively transit the pre-trained model from a simplistic probability distribution (e.g., softmax) to the more robust Dirichlet Distribution (evidence) via deep evidential learning. We provide extensive studies encompassing multiple scenarios, catering to ID and OOD image-text retrieval tasks. Specifically, the OOD retrieval widely covers different noisy settings, including simple noisy and web-style noisy images and text. The proposed methodology is subjected to rigorous theoretical scrutiny and empirical validation, substantiating its efficacy in achieving dependable image-text retrieval and accurate uncertainty estimation. The efficiency and scalability inherent in our approach render it well-suited for precise and expeditious uncertainty estimation within cross-modal systems, especially within domains that demand safety-critical predictions in the context of image-text alignment.

While this work improves the robustness of multi-modal alignment against web OOD cases, it is crucial to keep exploring noisy samples to enhance our method for better web applications. Specifically, in real-world scenarios, web data may include unintended private information, unsuitable images, or biased texts, leading to more complex OOD samples in a multi-modal context. We hope the proposed method could inspire future work to focus more on improving the robustness of vision-language modeling.

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
