# OpenReview forum: "Aligning Out-of-Distribution Web Images and Caption Semantics via Evidential Learning"
_ACM.org/TheWebConf/2024/Conference — TheWebConf24_

### Official Review · Reviewer_6jGU · 2023-11-14

**Novelty:** 6
**Technical Quality:** 6

**Review:**

This paper proposes an improvement over multi-modal retrieval models in order to better handle out-of-distribution (ood) images and text by using evidential learning. This model is evaluated using two well-known datasets against pre-trained models CLIP and BLIP. The authors claim that the evaluation shows that their model (ELIP) is more robust when handling OOD images and text, as there is a lower drop in recall.

Pros:
- The paper proposes a clear architectural improvement to SOTA models for cross-model retrieval
- The paper can be followed easily
- The experimental setting is appropriate

Cons:
- It is my impression that the MMI scores are not correctly computed (see detailed comments)
- It is my opinion that MMI is not the best way to quantify the improvement when using OOD text and images
- The text needs proper grammar check

Detailed comments:

My main concern is with the computation and use of MMI to evaluate and compare ELIP, CLIP and BLIP. Regarding the computation, from reference [23], I got that the formula to compute MMI is $(s_c-s_p)/s_c$, where $s_c$ is the "clean" score and $s_p$ is the perturbed score. In the case of the paper, these scores are R@1, R@5, and R@10. In the upper part of Table 1, the authors present the scores for the cases $T\rightarrow I$, $T\rightarrow I^*$, $T^*\rightarrow I$, and $T^*\rightarrow I^*$. My first question here is, which of the last three scenarios should be used to compute MMI against the scores of the "clean" scenario ($T\rightarrow I$). No mater which one is chosen, the numbers do not seem to be correct. Let's look closely to the MMI scores for ELIP in the upper portion of Table 1:

In the case of R@1, ELIP in the "clean" case has a score of 60.4. The "perturbed" R@1 scores are 51.5, 52.3, and 43.7 for the cases $T\rightarrow I^*$, $T^*\rightarrow I$, and $T^*\rightarrow I^*$ respectively. Therefore, computing MMI with the given formula, I get the following values when comparing the clean score against each of  the perturbed scores: 14.7%, 13.4%, and 27.6%. None of which correspond with the 18.6% figure shown in the table. I repeated this computation with the other R@k scores and none of my computations matched the ones shown in the paper. Later on I got that 18.6% is the average between the three numbers I got, so this should be explicit in the text.

Secondly, there is the matter of the use of the MMI metric to compare the models' ability to be robust against OOD text and image use. It is my opinion that this metric can benefit a model when it has a lower in-distribution performance, which is a contributing factor for a lower MMI. Therefore, it cannot be immediately concluded, just from seeing MMI scores, that a model handles better the OOD case. This is because if two models $m_1$ and $m_2$ have clean scores of $100$ and $90$ and perturbed scores of $80$ each, the MMI for $m_1$ is 20% and for $m_2$ is 11%, when arguably, $m_1$ is the better model. I also believe that this might be the reason for which [23] is not published in a journal or conference.

Regarding the noise introduction, some parameters should be discussed. For instance, the gaussian blur applied seems too aggressive in Fig. 2, so we should know the value of $\sigma$ that was used and why. Similarly for the text modifications.

In addition, I believe that the ELIP+ improvement over ELIP is never discussed in detail and simply just appears in page 6. It is also not said over which dataset the ablation study was conducted.

Finally, there are several writing issues that can be easily fixed (missing articles is the most common thing). Find in the following some detailed examples:

- Do not include cites or references in the abstract: [15,24] or Fig. 1
- Figure and Table captions are too long, detailed explanation should be given in the text
- Line 272: Adapter module [7] can be easily plugin-and-play into existing network -> An adapter module [7] can be easily plugged in into the existing network
- Line 312: the overall uncertainty mass of $u$ -> the overall uncertainty mass $u$
- Line 337: the prediction -> the predicted
- Line 340: similarity between query -> similarity between the query
- Line 378: cross-embeding -> cross-embeddings
- Line 507: be compute as -> be computed as
- Line 511: weights of $KL$ term -> weights of the $KL$ term
- Line 902: to effectively against -> to effectively what?
____________________________

After the authors response, my criticism on the use of MMI has been addressed by clarifying the figures shown in the paper and introducing a second performance metric. My other questions were answered, and the authors even went beyond them and added more disturbances to the images and provided me with the parameters they used and where they taken these from.

**Questions:**

- What is the explanation on the seeming mis-match of MMI computations?
- Why use MMI in the first place?
- What is ELIP+?
- What dataset was used for the ablation study?
- What are the parameters used for the noise-adding processes?

**Reviewer Confidence:**

3: The reviewer is confident but not certain that the evaluation is correct

**Scope:**

3: The work is somewhat relevant to the Web and to the track, and is of narrow interest to a sub-community

---

### Official Review · Reviewer_jcV3 · 2023-11-17

**Novelty:** 4
**Technical Quality:** 3

**Review:**

Vision-language models align disparate domains into a co-embedding space, showing impressive zero-shot performance on multi-modal tasks like image-text retrieval and visual question answering (VQA). The author argues existing methods often struggle with out-of-distribution (OOD) samples that exhibit noise and variability encountered in real-world scenarios, leading to significant performance drops. This paper introduces a framework called Evidential Language-Image Posterior (ELIP) to address this issue, leveraging evidential uncertainties to achieve robust alignment between web images and semantic knowledge across various OOD cases. The author claims ELIP can be integrated into general image-text contrastive learning frameworks, offering an efficient fine-tuning approach without requiring additional data. The effectiveness of ELIP is validated through testing on OOD cases, such as image distortion, spelling errors, and their combinations, on two benchmark datasets, simulating noisy data in real-world web applications.
Advantages: The paper is well-organized and easy to follow. The author intends to address the problem of OOD performance, which is a worthwhile topic. The framework is clearly presented with mathematical expressions.
Disadvantages: 1. To deal with the aforementioned problem,  this paper adopts evidential learning to enhance OOD performance. Although this is a classical approach to tackling such issues, It’s an integration of existing modules and the overall novelty of this framework is marginal.  2. Besides, there are lots of other methods, but the author did not present enough experiments comparing these counterparts. For instance, one potential way is to use ensemble methods by training multiple models and combining their predictions. Ensemble methods can often improve generalization and make the model more robust to different types of data. Introduce adversarial training by including examples that are intentionally designed to be challenging for the model. This can help the model become more robust to variations and outliers. 3. In the main experiments, the proposed framework is only compared with CLIP and BLIP. Also, the implementation of more datasets is expected.

**Questions:**

How does the proposed evidential learning approach compare to existing methods in terms of performance, robustness, and efficiency?

Are there clear advantages or limitations identified in the comparison? I would like to see more discussions and comparisons in the experimental session.

Since some noises are manually simulated, to what extent does the proposed method generalize to diverse datasets and real-world scenarios?

Are there insights into the model's transferability across different tasks and domains?

Given the computational demands associated with some probabilistic approaches, how does the proposed evidential learning method address issues related to scalability and efficiency?

**Reviewer Confidence:**

3: The reviewer is confident but not certain that the evaluation is correct

**Scope:**

3: The work is somewhat relevant to the Web and to the track, and is of narrow interest to a sub-community

---

### Official Review · Reviewer_j2CD · 2023-11-23

**Novelty:** 6
**Technical Quality:** 6

**Review:**

This paper tackles the problem of improving vision-language model robustness when handling noisy, out-of-distribution (OOD) image-text pairs commonly found in web applications. The proposed ELIP method outperforms SOTA vision-language models on OOD image-text retrieval, demonstrating improved robustness. Uncertainty estimates also reliably indicate OOD samples.

## Strengths
- Novel way of incorporating evidential learning to induce reliability and uncertainty awareness for pretrained VL models.
- Strong results on multiple OOD retrieval benchmarks and cases highlighting improved robustness over competitive baselines.

## Weaknesses
- Could experiment with wider range of real-world perturbations.

**Questions:**

- Has the author tried other more perturbation methods to generate the OOD dataset? A wider range of real-world perturbations will make the results of this article more more convincing.

**Reviewer Confidence:**

3: The reviewer is confident but not certain that the evaluation is correct

**Scope:**

4: The work is relevant to the Web and to the track, and is of broad interest to the community

---

### Official Review · Reviewer_G7DB · 2023-11-27

**Novelty:** 5
**Technical Quality:** 5

**Review:**

This paper first  comprehensively analyzes the performance drop caused by OOD factor. It introduces and designs multiple OOD cases to investigate large-scale VL models against various noise on web data. It proposes a novel uncertainty-aware, parameter-efficient tuning method termed ELIP. Extensive experiments show the effectiveness of the proposed method which improves state-of-the-art VL model.

**Questions:**

Since the ELIP+ is initialized by BLIP (line 569), why its performance is lower than BLIP in some settings (table 1). Could you provide a explanation?

**Reviewer Confidence:**

3: The reviewer is confident but not certain that the evaluation is correct

**Scope:**

4: The work is relevant to the Web and to the track, and is of broad interest to the community

---

### Official Review · Reviewer_rBGR · 2023-12-01

**Novelty:** 4
**Technical Quality:** 3

**Review:**

This work analyzes the performance drop between in-distribution (ID) and OOD retrieval.

Clearly missing evaluation with GPT-4V

**Questions:**

N/A

**Reviewer Confidence:**

4: The reviewer is certain that the evaluation is correct and very familiar with the relevant literature

**Scope:**

2: The connection to the Web is incidental, e.g., use of Web data or API

---

### Decision · Program_Chairs · 2024-01-22

**Decision:**

Accept

**Comment:**

The authors present a method, ELIP, to enhance the robustness of vision-language models in handling noisy, out-of-distribution (OOD) image-text pairs commonly encountered in web applications. The paper's strengths lie in a a architectural improvement proposed for cross-model retrieval and
 extensive experiments demonstrating the method's superiority over state-of-the-art vision-language models in OOD image-text retrieval, simulating real-world web application.

 Through the answer's to reviewers concerns the authors justify some of the weaknesses, such as comparisons primarily made (limited) against CLIP and BLIP, the lack of experimentation with other perturbation methods, queries about GPT-4V and MMI scores, etc. The authors respond effectively, addressing concerns by providing additional experimental results on more OOD cases and introducing a new evaluation metric (RSUM). The inclusion of an adapter ensemble approach for comparison is a positive addition.

 To the authors, please do make sure the changes discussed during the rebuttal, all the clarifications and new experiments are added in the updated version